# Response Rate of the Third and Fourth Doses of the BNT162b2 Vaccine Administered to Cancer Patients Undergoing Active Anti-Neoplastic Treatments

**DOI:** 10.3390/diseases11040128

**Published:** 2023-09-26

**Authors:** Abed Agbarya, Ina Sarel, Tomer Ziv-Baran, Orna Schwartz, Yelena Shechtman, Ella Kozlener, Rasha Khoury, Mohammad Sheikh-Ahmad, Leonard Saiegh, Forat Swaid, Asala Abu Ahmad, Urska Janzic, Ronen Brenner

**Affiliations:** 1Bnai-Zion Medical Center, Oncology Institute, Haifa 3339419, Israel; yelena.shechtman@b-zion.org.il (Y.S.); ella.kozlener@b-zion.org.il (E.K.); rasha.khoury@b-zion.org.il (R.K.); mohammad.ahmad@b-zion.org.il (M.S.-A.); leonard.saiegh@b-zion.org.il (L.S.); forat.swaid@b-zion.org.il (F.S.); asla.abu-ahmad@b-zion.org.il (A.A.A.); 2The Ruth and Bruce Rappaport Faculty of Medicine, Technion-Israel Institute of Technology, Haifa 3109601, Israel; 3Edith Wolfson Medical Center, Oncology Institute, Holon 5822012, Israel; Inasarel707@gmail.com; 4School of Public Health, Sackler Faculty of Medicine, Tel-Aviv University, Tel-Aviv 6997801, Israel; zivtome@tauex.tau.ac.il; 5Microbiology and Immunology Laboratory, Edith Wolfson Medical Center, Holon 5822012, Israel; ornas@wmc.gov.il; 6Department of Medical Oncology, University Clinic Golnik, 4202 Golnik, Slovenia; urska.janzic@klinika-golnik.si

**Keywords:** SARS-CoV-2, BNT162b2 vaccine, cancer patients, anti-neoplastic treatment, chemotherapy, antibodies, immunogenicity, COVID-19, immunocompromised

## Abstract

The BNT162b2 vaccine is globally used for preventing morbidity and mortality related to COVID-19. Cancer patients have had priority for receiving the vaccine due to their diminished immunity. This study reports the response rate of administering the third and fourth vaccine doses to cancer patients receiving active anti-neoplastic treatment. A total of 142 patients received two doses of the mRNA-based BNT162b2 COVID-19 vaccine, while 76 and 25 patients received three and four doses, respectively. The efficacy of the humoral response following two vaccine doses was diminished in cancer patients, especially in the group of patients receiving chemotherapy. In a multivariate analysis, patients who received three and four BNT162b2 vaccine doses were more likely to have antibody titers in the upper tertile compared to patients who received two doses of the vaccine (odds ratio (OR) 7.62 (95% CI 1.38–42.12), *p* = 0.02 and 17.15 (95% CI 5.01–58.7), *p* < 0.01, respectively). Unlike the response after two doses, the third and fourth BNT162b2 vaccine booster doses had an increased efficacy of 95–100% in cancer patients while undergoing active treatment. This result could be explained by different mechanisms including the development of memory B cells.

## 1. Introduction

After the global outbreak of severe acute respiratory syndrome coronavirus 2 (SARS-CoV-2) and the high number of deaths caused by the coronavirus disease (COVID-19) [1,2,3], a pandemic was declared on 11 March 2020 by the World Health Organization (WHO) [4]. Immune-compromised individuals have been more susceptible to the disease [5,6]. Cancer patients are at a higher risk for SARS-CoV-2 infection and have a higher mortality rate than the general population [7,8,9]. This is probably related to the low activity state of their immune system due to their cancer disease and the immunosuppressive treatments they receive [10,11]. The United States Food and Drug Administration (FDA) have urgently approved the SARS-CoV-2 vaccines, including the BNT162b2 (Pfizer-BioNTech, Mainz, Germany) and mRNA-1273 (Moderna Biotech, Cambridge, MA, USA) vaccines, to reduce the spread of infection and its severity [12,13].

While several trials have shown a relatively reduced immunogenicity for the BNT162b2 vaccine in patients with solid tumors receiving anti-neoplastic treatment [14,15,16], a recent prospective trial has shown that most patients develop an adequate antibody response to vaccination with the mRNA-1273 COVID-19 vaccine [17]. Most of these trials have tested the effect after one or two doses of the BNT162b2 vaccine [18].

In Israel, the BNT162b2 vaccine has been widely used [19,20], and cancer patients have had priority for receiving the vaccine [21]. We previously reported that after two doses of the BNT162b2 vaccine, a reduced immunogenicity was observed in this group of patients, especially in those receiving chemotherapy [22].

The aim of the present study is to assess the response of cancer patients undergoing anti-neoplastic treatments to the additional third and fourth doses of the mRNA SARS-CoV-2 vaccine.

The current study reports that administering the third and fourth doses of the anti-COVID-19 vaccine to a group of cancer patients undergoing anti-neoplastic treatments has a high efficacy, resulting in a 95–100% response rate demonstrated by all participants, who developed antibodies to COVID-19 following the fourth dose.

## 2. Materials and Methods

### 2.1. Study Design and Participants

The Institutional Review Boards of the Edith Wolfson Medical Center (WMC) and Bnai Zion Medical Center (BZMC) approved the study protocol. A written informed consent form was signed by all participants who were recruited at both medical centers between January 2021 and February 2022. Inclusion criteria were oncology patients ≥18 years old bearing solid tumors who were receiving active anti-neoplastic treatment when the second, third or fourth dose of the BNT162b2 vaccine (Pfizer-BioNTech, Kalamazoo, MI, USA) was administered. The third and fourth vaccine doses were administered six and three months (respectively) from the previous vaccine, while the second dose was delivered three weeks following the first vaccine dose. Participants who acquired COVID-19 or had a serological response indicating a past infection were excluded.

Demographic and clinical characteristics were collected through self-responded questionnaires filled by the patients in addition to detailed data collected from medical record files as previously described (Table 1) [22]. Among the patients recruited to this study were those harboring breast cancers, gastrointestinal cancers, gynecological cancers, lung cancers, melanoma, urinary tract cancers and other types. The cohort was sorted by type of treatment such as conventional cytotoxic chemotherapy vs. targeted therapies such as immunotherapy and biological agents. The treatment drugs were listed elsewhere [22].

### 2.2. Determination of Anti-SARS-CoV-2 Antibodies Level

COVID-19 vaccine triggers the immune system against the S-protein, which is detected through laboratory analysis. Anti-SARS-CoV-2 antibodies level was determined as previously described [22]. Peripheral venous blood samples were obtained from the participants at their clinic visit, which took place at least 7 days following the uptake of the second, third or fourth vaccine dose; the serum was separated and held at 4 °C until the analysis was performed. An ARCHITECT analyzer (Abbott, Abbott Park, IL, USA) was used for SARS-CoV-2 IgG II Quant assay at WMC Immunology laboratory. Quantification of IgG antibodies that are able to bind to the S1 subunit of the receptor-binding domain of the SARS-CoV-2 spike protein were detected via high-throughput chemiluminescent microparticle immunoassay technology. The assay threshold was 6.5 arbitrary units (AU) per mL; for International Standard, it was converted to Binding Antibody Units (BAU) by a factor of 0.142 (0.923 BAU/mL) and the maximal quantitation reached 5680 BAU/mL. IgG levels < 7.1 BAU/mL were considered negative, values between 7.1 BAU/mL and 21.3 BAU/mL were considered borderline and values >21.3 BAU/mL were included as positive results. Borderline scores were considered negative during the statistical analysis.

The serum samples from patients collected after three and four vaccine doses were also assessed on an ARCHITECT analyzer for the presence of SARS-CoV-2 IgG antibodies against the SARS-CoV-2 nucleocapside (anti-N). The assay was used to identify participants with an adaptive immune response to SARS-CoV-2, indicating recent or prior infection. Participants with COVID-19 Anti-N IgG (S/C index) above 1.4 were considered positive according to the manufacturer’s instructions and were excluded from the study.

### 2.3. Statistical Analysis

Categorical variables were summarized as frequency and percentage. Continuous variables were evaluated for normal distribution using histogram and box plot and reported as mean and Standard Deviation (SD) for normally distributed variables or as median and Interquartile Range (IQR) for skewed variables. Independent samples *t*-test or the Mann–Whitney test were used to compare continuous variables. Logistic regression calculations evaluated crude and Adjusted Odds Ratio. A multivariate analysis included parameters such as age, gender and days from last COVID-19 vaccine and chemotherapy. Two tailed statistical tests were performed. *p* < 0.05 was considered statistically significant. SPSS software was used for all the statistical analyses (IBM SPSS Statistics for Windows, version 27, IBM Corp., Armonk, NY, USA, 2020).

## 3. Results

### 3.1. Study Population

Two hundred and forty-three cancer patients undergoing active anti-neoplastic treatment were included in this study. A total of 142 patients received two doses of the BNT162b2 vaccine, 76 patients received three doses and 25 patients received four doses of the anti-SARS-CoV-2 vaccine. The patients’ demographic characteristics, cancer types and treatments are summarized in Table 1. The median age was 67 years (IQR 56.75–75), 66.5 years (IQR 57–74.75) and 72 years (IQR 67.5–79) in the second, third and fourth vaccine dose cohorts, respectively. Men represented 54.9%, 56.6% and 60% of the patients in the second, third and fourth vaccine dose cohorts. The highest incidence cancer types were gastrointestinal, breast and lung malignancies. The treatment regimens were administered via conventional cytotoxic chemotherapy agents or targeted therapies such as immunotherapy and biological agents.

### 3.2. Antibody Levels and Seropositivity Response to BNT162b2 Vaccine

The median SARS-CoV-2 IgG levels were significantly different (*p* < 0.001) among the groups: the median IgG levels following the second vaccine dose were 316 BAU/mL (IQR 63.2–1139.3), 571.6 BAU/mL (IQR 237.9–1119.5) and 3747 BAU/mL (IQR 176–5680) after the third and fourth BNT162b2 doses, respectively (Table 2 and Figure 1a). When the humoral response was stratified according to the treatment received, only after the second vaccine were there significant differences in the seropositivity (Figure 1b) and antibody levels (Figure 1a and Table 2) between the participants who received conventional cytotoxic chemotherapy and those who were administered targeted therapies (*p* < 0.001).

Analyzing the IgG response between the patients actively receiving conventional cytotoxic chemotherapy vs. non-chemotherapy treatment administered to patients treated with target therapies demonstrated no significant difference in the antibody levels (Table 2, and Figure 1a) (*p* = 0.443 and *p* = 0.81, respectively) and in seropositivity (Figure 1b) after the third and fourth vaccine doses.

No significant adverse effects of the vaccines were reported among the participating cancer patients (data not shown).

Figure 2 compares the effect of different variables between the patients after two, three or four vaccine doses.

After adjusting for gender, age and days from sample collection since the last COVID-19 vaccine and conventional cytotoxic chemotherapy treatment, the patients who received the third and fourth mRNA BNT162b2 vaccine doses had a significantly higher probability to have antibody titers in the upper tertile compared to the patients who received two doses of the vaccine (odds ratio (OR) 7.62 (95% CI 1.38–42.12), *p* = 0.02 and 17.15 (95% CI 5.01–58.7), *p* < 0.01, respectively). The titer of antibodies and their binding assay to specific SARS-CoV-2 depends on the individual, and usually, a high titer is correlated with a long persistence.

There was no significant difference between genders among the participants. The age, however, was different; the patients who received four doses of the vaccine were older than those who received two or three doses. Nevertheless, their serologic response was not impaired (Table 2).

Although the time since the last vaccine dose was significantly longer in the patients after three doses of vaccine, their serologic response was still positive.

## 4. Discussion

The COVID-19 outbreak has caused an unprecedented challenge for the medical system worldwide [23,24,25]. High morbidity rates among individuals with chronic conditions and senior individuals have been recorded [26,27]. Patients with conditions such as immune suppression were at higher risks for becoming infected with the SARS virus, complications and mortality [28]. Cancer patients have a higher risk and susceptibility to infectious diseases (especially respiratory system) due to a compromised immune system as a result of the chronic disease and the immune-suppressive treatments they receive [29]. Many studies have highlighted the importance of the vaccine for this group of patients [30,31]. Following the first two vaccine doses, the majority of cancer patients surveyed were found to carry antibodies against COVID-19, with diminished levels in patients receiving conventional cytotoxic chemotherapy [32,33].

We previously conducted a study aiming to analyze the humoral response of oncology patients diagnosed with solid tumors who were administered the first two BNT162b2 mRNA vaccine doses during their on-going anti-cancer treatment. It was compared to a matched group of participants who had not been diagnosed with cancer. The humoral response of the cancer patients undergoing active anti-neoplastic treatment was significantly reduced including seronegative results and lower antibody levels compared with that of the non-cancer individuals [22,34].

The present study analyzes the humoral response of cancer patients with solid malignancies following the third and fourth doses of the BNT162b2 vaccine.

Interestingly, 74 of 76 cancer patients were seropositive (97.4%) following the third dose of the anti-COVID-19 vaccine compared to 85.7% of cancer patients who were tested after the second vaccine dose in a previous study [22]. Similar results were also reported for solid organ transplant recipients, where 44% of the recipients who had been seronegative after two doses of the BNT162b2 vaccine became seropositive following the administration of a third vaccine dose [35]. The Moderna mRNA-1273 vaccine also resulted in a higher antibody titer in kidney transplant recipients who were borderline responsive after two doses [36].

Additionally, the recent study has shown no significant difference between the anti-cancer protocols, i.e., all participants who received the fourth BNT162b2 dose were seropositive regardless of their treatment regimen, including conventional cytotoxic chemotherapy vs. non-chemotherapy treatment administered to patients with targeted therapies. This may indicate that the patients’ immune systems could still respond well to the immunization. The path/process of producing IgG could be unharmed and work independently as a result of the immunization program without being affected by the active anti-cancer treatment.

Some explanations regarding these results can be postulated: First, cancer patients have a diminished immune response to vaccinations due to their underlying malignancy and anti-cancer treatment [37,38]. As a result, their humoral immunity can be boosted by additional doses of the BNT162b2 vaccine that can lead to a more robust immune response reflected by higher antibody levels [39,40,41,42,43].

Second, varied timing of the antibody sampling may represent different levels of antibody responses. A previous study was conducted 7 days after the second vaccination was administered [22], while in the present study, the blood collection was performed at a median of 140 days (IQR 130.5–160.8) and 18 days (IQR 12–25) following the third and fourth vaccine doses, respectively. The finding of such a high seropositive percentage of patients bearing antibodies to SARS-CoV-2 over 4 months following the third vaccine administration could point to the long duration of the response to this dose. The antibody levels may have been low immediately after the second dose, but increased progressively over time. Shroff et al. [44] evaluated the immune response after the third dose of the BNT162b2 vaccine in a phase 1 trial of 20 cancer cohort patients undergoing active anti-cancer therapy. Supporting the timely gradual development of antibodies, their report included two participants who were initially found to be seronegative by one week after the second vaccination; however, their follow-up monitoring detected antibodies before the third dose. Similar results were reported by Shmueli et al. [45]. These reports may postulate that some individuals bearing a malignant disease may have a slower response in producing antibodies.

Six months following the BNT162b2 vaccination, immunity in immunocompetent individuals reportedly started to decline [46,47,48,49]. A BNT162b2 booster dose can increase the antibody neutralization level by an average of five to seven times compared with that after a second dose [50,51,52]. Another study with 20 cancer patients, with solid tumors being actively treated, who were seronegative after the second BNT162b2 dose, showed 95% seropositivity after the administration of the third BNT162b2 vaccine dose [53].

In addition to humoral and T-cell-mediated immunity, memory B cells play a major role in seropositivity to COVID-19 vaccinations and are predictive of anamnestic responses after booster vaccination. Vietri et al. assumed that SARS-CoV-2-specific memory lymphocytes induced by the first vaccination trigger a faster and more effective antibody response following a booster vaccine [54,55,56,57,58,59]. Moreover, Shroff [44] detected the spike receptor-binding domain and other S1-specific memory B cell subsets as potential predictors of anamnestic responses to additional immunizations in most patients with cancer.

The third BNT162b2 vaccine dose was reported to elevate immunogenicity in bone marrow transplantation patients [60,61].

Healthcare workers vaccinated with two doses of mRNA 1273 in Belgium were observed to have a higher titer compared with the individuals vaccinated with BNT162b2 [62]. The Moderna mRNA 1273 COVID-19 vaccine was found in real-world data to be more effective than the Pfizer-BioNTech BNT162b2 vaccine (95% vs. 88%, respectively) in protecting immunocompromised individuals from hospitalization in the U.S.A following two doses [63]. A plausible reason was attributed to the higher concentration of the vaccine and the fact that the first two doses of the mRNA 1273 COVID-19 vaccine (100 μg and 50 μg, respectively) were administered 28 days apart, while the BNT162b2 vaccine (50 μg) was subcutaneously injected 21 days after the first vaccine. Oosting et al. [17] evaluated, in the Netherlands, the immunogenicity response of patients with solid tumors undergoing different treatment regimens to the mRNA-1273 vaccine (Moderna Biotech). Their findings indicate that the SARS-CoV-2 binding was of similar magnitude among all cancer patients after two doses, regardless of the treatment they received (chemotherapy, immunotherapy or combined chemoimmunotherapy). Moderna’s mRNA-1273 vaccine durability of a high response level, measured at 4 months following the second dose, was longer than the Pfizer’s BNT162b2 mRNA vaccine, which was waning down at 4–6 months’ time. In addition, a study of chronic lymphocytic leukemia and other non-Hodgkin lymphomas showed better antibody responses with the Moderna mRNA-1273 vaccine than with the Pizer-BioNTech BNT162b2 vaccine [64].

Hemodialysis immunosuppressed patients responded to the fourth vaccine (mRNA-1273) following three BNT162b vaccine doses, showing a strong augmentation of the humoral immunity against SARS-CoV-2 variants compared with the pre-vaccination level. These dialysis patients also had an increase in the T-cell responses as a result of the fourth vaccine [65].

Alexopopoulos et al. [66] reviewed effective clinical uses of interpreting the results of testing for antibodies against SARS-CoV-2. Following the BNT162B2 vaccine administration, the humoral response can be detected by analyzing the SARS-CoV-2 IgG of anti-spike IgG within two weeks. It is of utmost importance to be able to understand patients’ competence to develop antibodies, considering that the role of COVID-19 vaccination is to serve as prophylactic protection against a SARS infection. Alexopopoulos et al. suggested that antibody testing may evaluate the effect of immunosuppressive medications. According to the publication, the uptake of the third dose of the vaccine may provide better protection than the first and second doses against several variants such as omicron. Omicron mutants (BA.1), which evolved and spread during the fourth COVID-19 wave, vary from the original SARS-CoV-2 strains. Therefore, the vaccination through the first and second doses, which took place six months earlier, provided lower protection to this strain [66].

Immune checkpoint inhibitor treatment for patients bearing solid tumors was evaluated in relation to seroconversion by Terpos et al. [67]. It was demonstrated that the third dose following anti-COVID-19 vaccination boosted the antibody response in comparison with the first and second doses. That report supports the present study, which showed that after the third vaccine dose, the seropositivity was very high in addition to an enhanced humoral response following the third dose in patients with solid tumors.

Most cancer patients with hematological malignancies have low seroconversion rates (84.7%) vs. solid tumor patients (90.3%) after two doses [66].

The patients diagnosed with hematological malignancies while undergoing active treatment (including anti-CD20 therapy) were reported to have 76.3% of cases of developed humoral immunity. It was reported that among the cancer patients who received anti-CD 20 monoclonal antibody, the seroconversion rate following the anti-COVID vaccination was low for the six months following treatment. However, on a later date (one to two years), the calculated seroconversion rate increased, suggesting that the anti-CD 20 therapy caused a reduced humoral response [66].

Bergamaschi et al. [68] reported that in cancer patients who were immunocompromised due to therapy and had undergone bone marrow transplantation, their response to the first and second doses of the BNT162b2 mRNA vaccination yielded a low titer of the anti-S (SARS-CoV-2 Spike) antibodies to the Wuhan strain in comparison to the healthy control cohort. The analysis demonstrated that cytokines associated with inflammation were upregulated during vaccination. The authors concluded that the BNT162b2 mRNA vaccine induced cytokine changes that could serve as predictors for antibody titer development.

Rosati et al. [69] assessed Multiple Myeloma (MM) and Waldenstrom macroglobulinemia (WM) patient cohorts administered with the BNT162b2 mRNA vaccination. The observed increase in neutralizing the anti-spike antibody response following the third vaccine points to its benefit for MM and WM patients undergoing active treatment.

A review of 60 studies evaluating the COVID-19 vaccine efficacy in patients with either solid tumors or hematological malignances was conducted by Liatsou et al. [70]. The findings regarding humoral response showed that after the third vaccine dose, the effect size of the seroconversion rate for patients undergoing active therapy was estimated at 0.63 (95%CI: 0.54–0.72) for hematological cancers and 0.88 (95% CI: 0.75–0.97) for solid tumors. The cohort of patients with hematological malignancies were more affected in terms of anti-SARS-CoV-2 production due to the type of cancer and treatment with monoclonal antibodies.

In conclusion, oncology patients with hematological malignancies should be protected by vaccination to augment their immune response and any other available anti-viral drugs as a prophylaxis measure.

The limitations of the study include small cohorts and a non-uniform time used for sampling the participants. An important limitation of the current study could be attributed to the fact that it was not a longitudinal study, but rather a “snapshot”/“status” presentation [71] of the oncology patients’ antibody response to the COVID-19 vaccine while receiving active anti-neoplastic treatment at the time. This type of study design has been performed because in most cases, it is difficult to continue a follow-up of cancer patients while undergoing anti-neoplastic treatment for a long period for several reasons, including the fact that some of the patients died or stopped treatment due to the fact they were cured, or sometimes, the decision of symptomatic treatment was only due to general deterioration and end-stage disease or was lost from the follow-up. Yet, Figure 1a and Table 2 indicate that the antibody response to the fourth vaccine dose was higher than for the third vaccine dose despite the fact that this is not a longitudinal study. Future research directions may incorporate an expanded cohort size and explore additional mechanisms involved in the immunity to SARS-CoV-2 following vaccination. It is important to note that by the time the third and fourth anti-COVID-19 vaccines were introduced, different virus variants of concern were spreading the infection in Israel and worldwide. Therefore, the production of the vaccine by pharmaceutical companies such as Pfizer (BNT162b2 mRNA 50 μg) has been adapted and updated accordingly to be specific to the delta/omicron strains. Furthermore, studying the response to the third and fourth doses of the Moderna mRNA-1273 50 μg anti-COVID-19 vaccine could be beneficial, as apparently, these mRNA vaccines were providing better protection to cancer patients.

## 5. Conclusions

In contrast to the second dose of the anti-COVID-19 vaccine, the third and fourth doses administered to cancer patients undergoing anti-neoplastic treatments had a high humoral response in chemotherapy-treated as well as in non-chemotherapy-treated patients.

Distinct pathways contribute to vaccine immunity with different measurable tests such as humoral immunity and T cell activity. While these tests provide valuable information, they are not the only indicators of immunity, and the vast majority of cancer patients have some level of protection against COVID-9 even if their antibody or T-cell levels are not as high as expected. The titer of antibodies and their binding assay to specific SARS-CoV-2 strains depends on each individual person, and usually, a high titer is correlated with a long persistence.

## Figures and Tables

**Figure 1 diseases-11-00128-f001:**
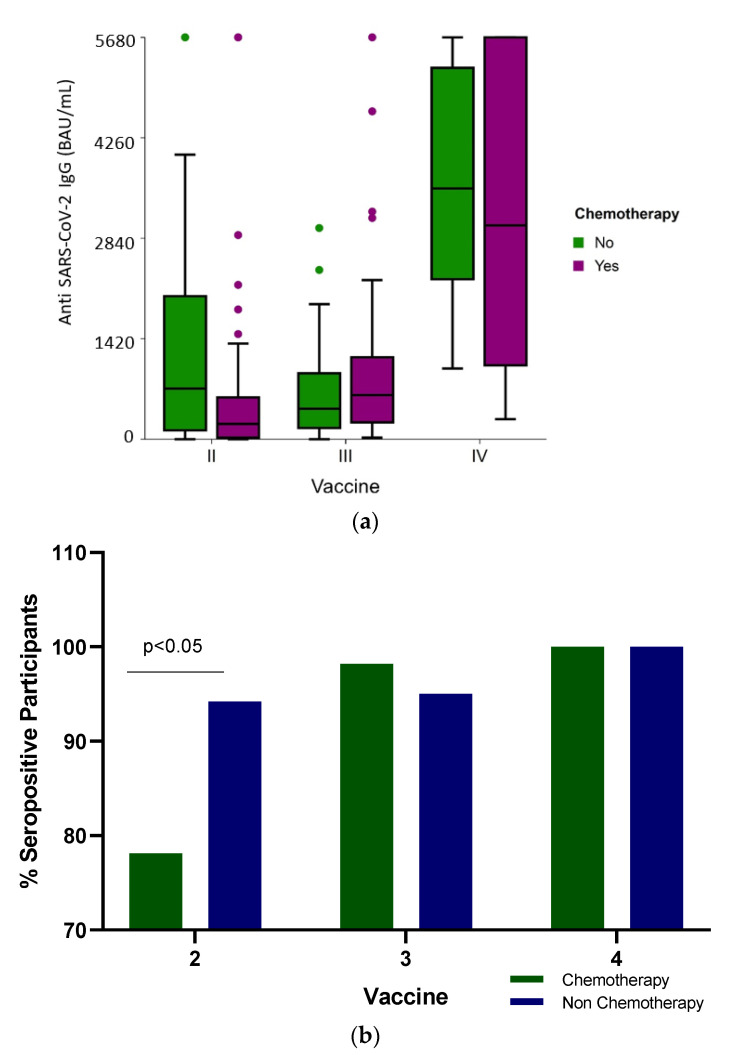
(**a**) Anti-SARS-CoV-2 antibody level (BAU /mL) distribution shows IgG titer following two, three and four BNT162b2 vaccine doses in cancer patients actively receiving conventional cytotoxic chemotherapy or non-chemotherapy treatment administered as targeted therapies. The boxplot includes the median and quartiles as horizontal lines. After two vaccine doses, the chemotherapy group of patients had low IgG scores. The data points outside the boxes represent outliers. (**b**) Vaccine response. Percent of seropositive cancer patients actively treated with chemotherapy or non-chemotherapy drugs following two, three and four BNT162b2 vaccine doses.

**Figure 2 diseases-11-00128-f002:**
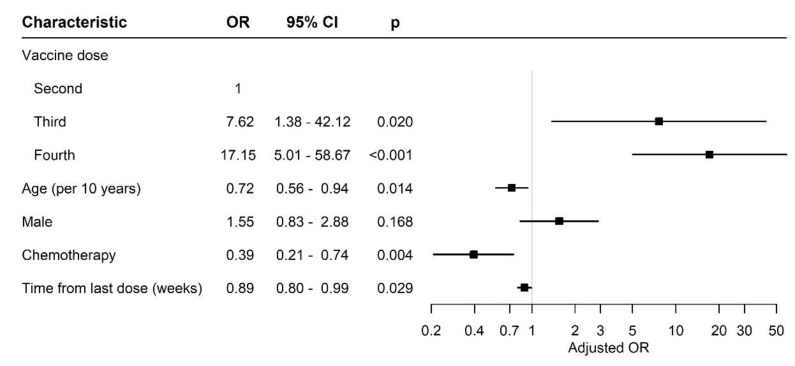
Forest plot of odds ratios. Abbreviations: OR, odds ratio; CI, confidence interval.

**Table 1 diseases-11-00128-t001:** Demographic and clinical characteristics of participants receiving BNT162b2 vaccine dose.

	After 2nd Dose	After 3rd Dose	After 4th Dose
Frequency (n)	142	76	25
Age, median years (IQR)	67 (56.75–75)	66.5 (57–74.75)	72 (67.5–79)
Male (%)	78 (54.9)	43 (56.6)	15 (60)
Time from last BNT162b2 dose, median days (IQR)	35 (24.5–46.25)	147 (130.5–160.8)	18 (12–25)
**Type of Cancer, n (%)**			
Gastrointestinal	49 (34.5)	42 (55.3)	10 (40)
Breast	30 (21.1)	12 (15.8)	3 (12)
Lung	28 (19.7)	10 (22.7)	6 (24)
Urinary	13 (9.2)	3 (3.9)	2 (8)
Melanoma	7 (4.9)	4 (5.3)	3 (12)
Gynecological	9 (6.3)	2 (2.6)	0
Other types	6 (4.2)	3 (3.94)	1 (4)
**Treatment, n (%)**			
Chemotherapy ^1^	73 (51.4)	56 (73.7)	13 (52)
Non-chemotherapy ^2^	69 (48.6)	20 (26.3)	12 (48)

^1^ Chemotherapy treatment administered to patients as conventional cytotoxic chemotherapy. ^2^ Non-chemotherapy treatment administered to patients treated with targeted therapies.

**Table 2 diseases-11-00128-t002:** Antibody response rate and titers after a second, third and fourth dose of the BNT162b2 mRNA vaccine in cancer patients.

	After 2nd	Vaccine	After 3rd	Vaccine	After 4th	Vaccine
	Chemo ^1^	Non-Chemo ^2^	Chemo	Non-Chemo	Chemo	Non-Chemo
Frequency (*n*)	73	69	56	20	13	12
COVID-19 antibodies response, *n* (%)						
Positive (>21.3 BAU)	57 (78.1)	65	55 (98.2)	19 (95)	13 (100)	12 (100)
Borderline (>7.1 BAU, <21.3 BAU)	4 (5.5)	2 (2.9)	1 (1.8)	0	0	0
Negative	12 (16.4)	2 (2.9)	0	1 (5)	0	0
Anti-SARS-CoV-2 IgG (BAU/mL), median (IQR)	316.8	(63.2–1139.3)	571.6	(237.9–1119.5)	3474	(176.8–5680)
R2A3						
after each vaccine dose	216.4 (25.3–593.2)	716.4 (121.5–2023.5)	620 (214.3–1153.9)	434.6 (61.76–928.02)	3019.2 (1043.3–5680)	3544.5 (2261.3–5243.6)
Antibody levels comparison chemo/non-chemo (*p*) for each vaccine	**<0.001**		**0.443**		**0.81**	

^1^ Chemo denotes chemotherapy treatment administered to patients as conventional cytotoxic chemotherapy. ^2^ Non-chemo denotes non-chemotherapy treatment administered to patients treated with targeted therapies.

## Data Availability

The data presented in this study are available upon request from the corresponding authors.

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
