# Peer review of "Response Rate of the Third and Fourth Doses of the BNT162b2 Vaccine Administered to Cancer Patients Undergoing Active Anti-Neoplastic Treatments"

_diseases, 2023, doi:10.3390/diseases11040128_

Round 1

Reviewer 1 Report (Previous Reviewer 2)

This version has explained all the concerns. 

Author Response

Please find attached the reply file 

Reviewer 2 Report (New Reviewer)

Major concerns.

1. Were participants with a history of COVID-19 infection or suspected before study enrolment excluded?

People with a history of COVID-19 may have a stronger immune response after a booster shot(s) than people who have never gotten COVID-19 before.
If this study included the outcome from participants with a history of COVID-19, the outcome could be distorted.

Suggest clarify this point.

2. Lines 97-98: "Briefly, at least 7 days after the administration of the second, third or fourth vaccine dose, peripheral venous blood samples were obtained...".

The neutralising antibody has a peak within 14 days after vaccination.  Why did this study collect blood 7 days after a recent vaccination?

Minor concerns.

1. Your immunologic assessment tool (SARS-CoV-2 IgG II Quant) can convert to BAU/mL by a conversion factor of 0.142. (BAU/mL = AU/mL x 0.142)
I suggest converting from AU/mL to BAU/mL to make it intercomparison with other validated immunologic assessment tools. It also describes the conversion factor in the section "2.2. Determination of Anti-SARS-CoV-2 Antibodies level ".

2. Lines 362-364 suggest adding more information about the dosage of each vaccine to make it clear why Spikevax is given more protection than Comirnaty.

The common dosage used for each vaccine is 1) BNT162b2 50 µg, 2) mRNA-1273 100 µg (as prime shot), and 3) mRNA-1273 50 µg (as booster shot).

Moreover, suggests using a research name (e.g. mRNA-1273) instead of a manufacturer name (e.g. Moderna) to make it consistent with the previously mentioned in the Introduction part.

Comments.

1. Suggest adding "COVID-19" and "immunocompromiz(s)ed" (depending on your English style) to the keywords to increase search visibility.

2. Line 114 suggests using "Anti-N" instead of "Anti N".

3. Line 122 suggests using "t-Test" instead of "t Test".

Suggest checking hyphenation and typos.

Author Response

To:

Ms. Vanessa Hu

Assistant Editor

 Diseases

[email protected]

                                                                                                September 18, 2023     

 Re: Diseases-2597313  re-submission  

Dear Ms. Hu,

Thank you for considering our manuscript "Response rate to the 3rd and 4th doses of BNT162b2 vaccine in cancer patients administered with active anti-neoplastic treatments" for publication in Diseases.

We thank the reviewers for their thoughtful comments and their helpful constructive feedback, giving us the opportunity to improve and revise the paper.

The reviewers' recommendations have now been incorporated into the manuscript and are documented in the following revision note.

Enclosed please find point by point responses to the referees' comments and the revised manuscript with revisions highlighted in light blue as additions and in yellow for deletions /REVIEW MODE. In the attached letter, R represents reviewer, Q denotes a question/comment by the reviewer, A presents an answer by the authors.

We thank you for the duplication and report. We have rewritten the parts needed revision.

We hope that the revised manuscript will fulfill the reviewers' comments to satisfaction.

Sincerely,

Dr. Abed Agbarya

Review Report Form 2

Comments and suggestions for Authors

Major concerns.

R2Q1. Were participants with a history of COVID-19 infection or suspected before study enrolment excluded?

People with a history of COVID-19 may have stronger immune response after a booster shot(s) than people who have never gotten COVID-19 before.

If this study included the outcome form participants with a history of COVID-19, the outcome could be distorted.

Suggest clarify this point.

R2A1. Yes, the exclusion criteria were documented in the previous version of this manuscript, these participants were addressed as follows:

 in section 2. Materials and Methods

subsection  2.1. Study Design and Participants

 lines 78-79 "Participants who acquired COVID-19 or had a serological response indicating a past infection, were excluded."

In addition, in subsection

2.2. Determination of Anti-SARS-CoV-2 Antibodies level

Lines 110-116

"The serum samples from patients collected after three and four vaccine doses were also assessed for the presence of antibodies against the SARS-CoV-2 nucleocapside (anti-N) using the SARS-CoV-2 IgG on an ARCHITECT analyzer. The assay was used to identify participants with an adaptive immune response to SARS-CoV-2, indicating recent or prior infection. Participants with COVID-19 Anti N IgG (S/C index) above 1.4 were considered positive according to the manufacturer's instructions, and excluded from the study "

R2Q2. Lines 97-98: "Briefly, at least 7 days after the administration of the second, third or fourth vaccine dose, peripheral venous blood samples were obtained…".

The neutralizing antibody has a peak within 14 days after vaccination. Why did this study collect blood 7 days after a recent vaccination?

R2A2

 The blood collection was done during the patients scheduled clinic appointments (such as on treatment day), which were at least 7 days following vaccination. 

Minor concerns.

R2Q3

Your immunologic assessment tool (SARS-CoV-2 IgG II Quant) can convert to BAU/mL by a conversion factor of 0.142. (BAU/mL = AU/mL x 0.142)
I suggest converting from AU/mL to BAU/mL to make it intercomparison with other validated immunologic assessment tools. It also describes the conversion factor in the section "2.2. Determination of Anti-SARS-CoV-2 Antibodies level ".

R2A3

The revised Table 2 is presented as follows

Table 2.  Antibody response rate and titers after a second, third and fourth dose of the BNT162b2 mRNA vaccine in cancer patients.

After 2nd

vaccine

After 3rd

vaccine

After 4th

vaccine

Chemo1

Non-Chemo2

Chemo

Non-Chemo

Chemo

Non-Chemo

Number (n)

73

69

56

20

13

12

COVID-19 antibodies response,

n (%)

Positive (>21.3 BAU)

57 (78.1)

65

55 (98.2)

19 (95)

13 (100)

12 (100)

Borderline (>7.1 BAU, <21.3 BAU)

4 (5.5)

2 (2.9)

1 (1.8)

0

0

0

Negative

12 (16.4)

2 (2.9)

0

1 (5)

0

0

Anti SARS-COV-2 IgG (BAU/mL), median (IQR)

316.8

(63.2-1139.3)

571.6

(237.9-1119.5)

3474

(176.8-5680)

R2A3

after each vaccine dose

216.4

(25.3-593.2)

716.4

(121.5-2023.5)

620

(214.3-1153.9)

434.6

(61.76-928.02)

3019.2

1043.3-5680)

3544.5

(2261.3-5243.6)

Antibodies levels comparison chemo/non chemo (p) for each vaccine

<0.001

0.443

0.81

1 Chemo denotes chemotherapy treatment administered to patients as conventional cytotoxic chemotherapy.

2 Non-Chemo denotes non-chemotherapy treatment administered to patients treated by targeted therapies.

As suggested by the reviewer, a description of the conversion factor was added to the section

2.2. Determination of Anti-SARS-CoV-2 Antibodies level

Lines 106-110

The assay threshold was 6.5 arbitrary units (AU) per mL, for International Standard it was converted to Binding Antibody Units (BAU) by a factor of 0.142 (0.923 BAU/mL); and the maximal quantitation reached 5680 BAU /mL. IgG levels < 7.1 BAU/mL were considered negative, values between 7.1 BAU/mL and 21.3 BAU/mL were considered borderline, values > 21.3 BAU/mL were included as positive results

Accordingly, lines 147-148 were corrected

3.2. Antibody levels and seropositivity response to BNT162b2 vaccine

The median SARS-CoV-2 IgG levels were significantly different (p< 0.001) among the groups: median IgG levels following the second vaccine dose were 316 BAU AU/mL (IQR 445-8023 63.2-1139.3); 4025 AU/ 571.6 BAU/mL (IQR 1676-7884 237.9-1119.5) and 24465 AU 3747 BAU/ml (IQR 1245-40000 176-5680) after the third and fourth BNT162b2 doses, respectively (Table 2 and Figure 1a).

  Figure 1a was updated as well.

R2Q4

Lines 362-364 suggest adding more information about the dosage of each vaccine to make it clear why Spikevax is given more protection than Comirnaty.

The common dosage used for each vaccine is 1) BNT162b2 50μg, 2) mRNA-1273 100 μg (as prime shot), and 3) mRNA-1273 50 μg (as booster shot).

Moreover, suggests using a research name (e.g. mRNA-1273) instead of a manufacturer name (e.g. Moderna) to make it consistent with the previously mentioned in the introduction part.

R2A4 We thank the reviewer for the suggestion and incorporated it into the text as follows:

"Therefore, the production of the vaccine by the pharmaceutical companies such as Pfizer BNT162b2 mRNA 50 μg (produced by Pfizer) have been adapted and updated accordingly, to be specific to delta / omicron strains. Furthermore, studying the response to Moderna mRNA-1273 50 μg (produced by Moderna) third and fourth anti-COVID-19 vaccine doses could be beneficial as apparently these mRNA vaccines were providing better protection to cancer patients. "

R2Q5 Comments.

  1. Suggest adding "COVID-19" and "immunocompromised" to the keywords to increase visibility

R2A5 COVID-19 and immunocompromised were added to the keywords, line 39:

Keywords SARS-CoV-2; BNT162b2 vaccine; cancer patients; anti-neoplastic treatment; chemotherapy; antibodies; immunogenicity; COVID-19; immunocompromised

R2Q6 Comments.

  1. Line 114 suggests using "Anti-N" instead of "Anti N".

R2A6 Anti-N replaces Anti N as follows

"Participants with COVID-19 Anti-N IgG (S/C index) above 1.4 were…"

R2Q7 Comments.

  1. Line 122 suggests using "t-Test" instead of "t Test"

R2A7 t-Test replaces t Test

"…using Independent Samples t-Test…"

To:

Ms. Vanessa Hu

Assistant Editor

 Diseases

[email protected]

                                                                                                September 18, 2023     

 Re: Diseases-2597313  re-submission  

Dear Ms. Hu,

Thank you for considering our manuscript "Response rate to the 3rd and 4th doses of BNT162b2 vaccine in cancer patients administered with active anti-neoplastic treatments" for publication in Diseases.

We thank the reviewers for their thoughtful comments and their helpful constructive feedback, giving us the opportunity to improve and revise the paper.

The reviewers' recommendations have now been incorporated into the manuscript and are documented in the following revision note.

Enclosed please find point by point responses to the referees' comments and the revised manuscript with revisions highlighted in light blue as additions and in yellow for deletions /REVIEW MODE. In the attached letter, R represents reviewer, Q denotes a question/comment by the reviewer, A presents an answer by the authors.

We thank you for the duplication and report. We have rewritten the parts needed revision.

We hope that the revised manuscript will fulfill the reviewers' comments to satisfaction.

Sincerely,

Dr. Abed Agbarya

Review Report Form 2

Comments and suggestions for Authors

Major concerns.

R2Q1. Were participants with a history of COVID-19 infection or suspected before study enrolment excluded?

People with a history of COVID-19 may have stronger immune response after a booster shot(s) than people who have never gotten COVID-19 before.

If this study included the outcome form participants with a history of COVID-19, the outcome could be distorted.

Suggest clarify this point.

R2A1. Yes, the exclusion criteria were documented in the previous version of this manuscript, these participants were addressed as follows:

 in section 2. Materials and Methods

subsection  2.1. Study Design and Participants

 lines 78-79 "Participants who acquired COVID-19 or had a serological response indicating a past infection, were excluded."

In addition, in subsection

2.2. Determination of Anti-SARS-CoV-2 Antibodies level

Lines 110-116

"The serum samples from patients collected after three and four vaccine doses were also assessed for the presence of antibodies against the SARS-CoV-2 nucleocapside (anti-N) using the SARS-CoV-2 IgG on an ARCHITECT analyzer. The assay was used to identify participants with an adaptive immune response to SARS-CoV-2, indicating recent or prior infection. Participants with COVID-19 Anti N IgG (S/C index) above 1.4 were considered positive according to the manufacturer's instructions, and excluded from the study "

R2Q2. Lines 97-98: "Briefly, at least 7 days after the administration of the second, third or fourth vaccine dose, peripheral venous blood samples were obtained…".

The neutralizing antibody has a peak within 14 days after vaccination. Why did this study collect blood 7 days after a recent vaccination?

R2A2

 The blood collection was done during the patients scheduled clinic appointments (such as on treatment day), which were at least 7 days following vaccination. 

Minor concerns.

R2Q3

Your immunologic assessment tool (SARS-CoV-2 IgG II Quant) can convert to BAU/mL by a conversion factor of 0.142. (BAU/mL = AU/mL x 0.142)
I suggest converting from AU/mL to BAU/mL to make it intercomparison with other validated immunologic assessment tools. It also describes the conversion factor in the section "2.2. Determination of Anti-SARS-CoV-2 Antibodies level ".

R2A3

The revised Table 2 is presented as follows

Table 2.  Antibody response rate and titers after a second, third and fourth dose of the BNT162b2 mRNA vaccine in cancer patients.

After 2nd

vaccine

After 3rd

vaccine

After 4th

vaccine

Chemo1

Non-Chemo2

Chemo

Non-Chemo

Chemo

Non-Chemo

Number (n)

73

69

56

20

13

12

COVID-19 antibodies response,

n (%)

Positive (>21.3 BAU)

57 (78.1)

65

55 (98.2)

19 (95)

13 (100)

12 (100)

Borderline (>7.1 BAU, <21.3 BAU)

4 (5.5)

2 (2.9)

1 (1.8)

0

0

0

Negative

12 (16.4)

2 (2.9)

0

1 (5)

0

0

Anti SARS-COV-2 IgG (BAU/mL), median (IQR)

316.8

(63.2-1139.3)

571.6

(237.9-1119.5)

3474

(176.8-5680)

R2A3

after each vaccine dose

216.4

(25.3-593.2)

716.4

(121.5-2023.5)

620

(214.3-1153.9)

434.6

(61.76-928.02)

3019.2

1043.3-5680)

3544.5

(2261.3-5243.6)

Antibodies levels comparison chemo/non chemo (p) for each vaccine

<0.001

0.443

0.81

1 Chemo denotes chemotherapy treatment administered to patients as conventional cytotoxic chemotherapy.

2 Non-Chemo denotes non-chemotherapy treatment administered to patients treated by targeted therapies.

As suggested by the reviewer, a description of the conversion factor was added to the section

2.2. Determination of Anti-SARS-CoV-2 Antibodies level

Lines 106-110

The assay threshold was 6.5 arbitrary units (AU) per mL, for International Standard it was converted to Binding Antibody Units (BAU) by a factor of 0.142 (0.923 BAU/mL); and the maximal quantitation reached 5680 BAU /mL. IgG levels < 7.1 BAU/mL were considered negative, values between 7.1 BAU/mL and 21.3 BAU/mL were considered borderline, values > 21.3 BAU/mL were included as positive results

Accordingly, lines 147-148 were corrected

3.2. Antibody levels and seropositivity response to BNT162b2 vaccine

The median SARS-CoV-2 IgG levels were significantly different (p< 0.001) among the groups: median IgG levels following the second vaccine dose were 316 BAU AU/mL (IQR 445-8023 63.2-1139.3); 4025 AU/ 571.6 BAU/mL (IQR 1676-7884 237.9-1119.5) and 24465 AU 3747 BAU/ml (IQR 1245-40000 176-5680) after the third and fourth BNT162b2 doses, respectively (Table 2 and Figure 1a).

  Figure 1a was updated as well.

R2Q4

Lines 362-364 suggest adding more information about the dosage of each vaccine to make it clear why Spikevax is given more protection than Comirnaty.

The common dosage used for each vaccine is 1) BNT162b2 50μg, 2) mRNA-1273 100 μg (as prime shot), and 3) mRNA-1273 50 μg (as booster shot).

Moreover, suggests using a research name (e.g. mRNA-1273) instead of a manufacturer name (e.g. Moderna) to make it consistent with the previously mentioned in the introduction part.

R2A4 We thank the reviewer for the suggestion and incorporated it into the text as follows:

"Therefore, the production of the vaccine by the pharmaceutical companies such as Pfizer BNT162b2 mRNA 50 μg (produced by Pfizer) have been adapted and updated accordingly, to be specific to delta / omicron strains. Furthermore, studying the response to Moderna mRNA-1273 50 μg (produced by Moderna) third and fourth anti-COVID-19 vaccine doses could be beneficial as apparently these mRNA vaccines were providing better protection to cancer patients. "

R2Q5 Comments.

  1. Suggest adding "COVID-19" and "immunocompromised" to the keywords to increase visibility

R2A5 COVID-19 and immunocompromised were added to the keywords, line 39:

Keywords SARS-CoV-2; BNT162b2 vaccine; cancer patients; anti-neoplastic treatment; chemotherapy; antibodies; immunogenicity; COVID-19; immunocompromised

R2Q6 Comments.

  1. Line 114 suggests using "Anti-N" instead of "Anti N".

R2A6 Anti-N replaces Anti N as follows

"Participants with COVID-19 Anti-N IgG (S/C index) above 1.4 were…"

R2Q7 Comments.

  1. Line 122 suggests using "t-Test" instead of "t Test"

R2A7 t-Test replaces t Test

"…using Independent Samples t-Test…"

To:

Ms. Vanessa Hu

Assistant Editor

 Diseases

[email protected]

                                                                                                September 18, 2023     

 Re: Diseases-2597313  re-submission  

Dear Ms. Hu,

Thank you for considering our manuscript "Response rate to the 3rd and 4th doses of BNT162b2 vaccine in cancer patients administered with active anti-neoplastic treatments" for publication in Diseases.

We thank the reviewers for their thoughtful comments and their helpful constructive feedback, giving us the opportunity to improve and revise the paper.

The reviewers' recommendations have now been incorporated into the manuscript and are documented in the following revision note.

Enclosed please find point by point responses to the referees' comments and the revised manuscript with revisions highlighted in light blue as additions and in yellow for deletions /REVIEW MODE. In the attached letter, R represents reviewer, Q denotes a question/comment by the reviewer, A presents an answer by the authors.

We thank you for the duplication and report. We have rewritten the parts needed revision.

We hope that the revised manuscript will fulfill the reviewers' comments to satisfaction.

Sincerely,

Dr. Abed Agbarya

Review Report Form 2

Comments and suggestions for Authors

Major concerns.

R2Q1. Were participants with a history of COVID-19 infection or suspected before study enrolment excluded?

People with a history of COVID-19 may have stronger immune response after a booster shot(s) than people who have never gotten COVID-19 before.

If this study included the outcome form participants with a history of COVID-19, the outcome could be distorted.

Suggest clarify this point.

R2A1. Yes, the exclusion criteria were documented in the previous version of this manuscript, these participants were addressed as follows:

 in section 2. Materials and Methods

subsection  2.1. Study Design and Participants

 lines 78-79 "Participants who acquired COVID-19 or had a serological response indicating a past infection, were excluded."

In addition, in subsection

2.2. Determination of Anti-SARS-CoV-2 Antibodies level

Lines 110-116

"The serum samples from patients collected after three and four vaccine doses were also assessed for the presence of antibodies against the SARS-CoV-2 nucleocapside (anti-N) using the SARS-CoV-2 IgG on an ARCHITECT analyzer. The assay was used to identify participants with an adaptive immune response to SARS-CoV-2, indicating recent or prior infection. Participants with COVID-19 Anti N IgG (S/C index) above 1.4 were considered positive according to the manufacturer's instructions, and excluded from the study "

R2Q2. Lines 97-98: "Briefly, at least 7 days after the administration of the second, third or fourth vaccine dose, peripheral venous blood samples were obtained…".

The neutralizing antibody has a peak within 14 days after vaccination. Why did this study collect blood 7 days after a recent vaccination?

R2A2

 The blood collection was done during the patients scheduled clinic appointments (such as on treatment day), which were at least 7 days following vaccination. 

Minor concerns.

R2Q3

Your immunologic assessment tool (SARS-CoV-2 IgG II Quant) can convert to BAU/mL by a conversion factor of 0.142. (BAU/mL = AU/mL x 0.142)
I suggest converting from AU/mL to BAU/mL to make it intercomparison with other validated immunologic assessment tools. It also describes the conversion factor in the section "2.2. Determination of Anti-SARS-CoV-2 Antibodies level ".

R2A3

The revised Table 2 is presented as follows

Table 2.  Antibody response rate and titers after a second, third and fourth dose of the BNT162b2 mRNA vaccine in cancer patients.

After 2nd

vaccine

After 3rd

vaccine

After 4th

vaccine

Chemo1

Non-Chemo2

Chemo

Non-Chemo

Chemo

Non-Chemo

Number (n)

73

69

56

20

13

12

COVID-19 antibodies response,

n (%)

Positive (>21.3 BAU)

57 (78.1)

65

55 (98.2)

19 (95)

13 (100)

12 (100)

Borderline (>7.1 BAU, <21.3 BAU)

4 (5.5)

2 (2.9)

1 (1.8)

0

0

0

Negative

12 (16.4)

2 (2.9)

0

1 (5)

0

0

Anti SARS-COV-2 IgG (BAU/mL), median (IQR)

316.8

(63.2-1139.3)

571.6

(237.9-1119.5)

3474

(176.8-5680)

R2A3

after each vaccine dose

216.4

(25.3-593.2)

716.4

(121.5-2023.5)

620

(214.3-1153.9)

434.6

(61.76-928.02)

3019.2

1043.3-5680)

3544.5

(2261.3-5243.6)

Antibodies levels comparison chemo/non chemo (p) for each vaccine

<0.001

0.443

0.81

1 Chemo denotes chemotherapy treatment administered to patients as conventional cytotoxic chemotherapy.

2 Non-Chemo denotes non-chemotherapy treatment administered to patients treated by targeted therapies.

As suggested by the reviewer, a description of the conversion factor was added to the section

2.2. Determination of Anti-SARS-CoV-2 Antibodies level

Lines 106-110

The assay threshold was 6.5 arbitrary units (AU) per mL, for International Standard it was converted to Binding Antibody Units (BAU) by a factor of 0.142 (0.923 BAU/mL); and the maximal quantitation reached 5680 BAU /mL. IgG levels < 7.1 BAU/mL were considered negative, values between 7.1 BAU/mL and 21.3 BAU/mL were considered borderline, values > 21.3 BAU/mL were included as positive results

Accordingly, lines 147-148 were corrected

3.2. Antibody levels and seropositivity response to BNT162b2 vaccine

The median SARS-CoV-2 IgG levels were significantly different (p< 0.001) among the groups: median IgG levels following the second vaccine dose were 316 BAU AU/mL (IQR 445-8023 63.2-1139.3); 4025 AU/ 571.6 BAU/mL (IQR 1676-7884 237.9-1119.5) and 24465 AU 3747 BAU/ml (IQR 1245-40000 176-5680) after the third and fourth BNT162b2 doses, respectively (Table 2 and Figure 1a).

  Figure 1a was updated as well.

R2Q4

Lines 362-364 suggest adding more information about the dosage of each vaccine to make it clear why Spikevax is given more protection than Comirnaty.

The common dosage used for each vaccine is 1) BNT162b2 50μg, 2) mRNA-1273 100 μg (as prime shot), and 3) mRNA-1273 50 μg (as booster shot).

Moreover, suggests using a research name (e.g. mRNA-1273) instead of a manufacturer name (e.g. Moderna) to make it consistent with the previously mentioned in the introduction part.

R2A4 We thank the reviewer for the suggestion and incorporated it into the text as follows:

"Therefore, the production of the vaccine by the pharmaceutical companies such as Pfizer BNT162b2 mRNA 50 μg (produced by Pfizer) have been adapted and updated accordingly, to be specific to delta / omicron strains. Furthermore, studying the response to Moderna mRNA-1273 50 μg (produced by Moderna) third and fourth anti-COVID-19 vaccine doses could be beneficial as apparently these mRNA vaccines were providing better protection to cancer patients. "

R2Q5 Comments.

  1. Suggest adding "COVID-19" and "immunocompromised" to the keywords to increase visibility

R2A5 COVID-19 and immunocompromised were added to the keywords, line 39:

Keywords SARS-CoV-2; BNT162b2 vaccine; cancer patients; anti-neoplastic treatment; chemotherapy; antibodies; immunogenicity; COVID-19; immunocompromised

R2Q6 Comments.

  1. Line 114 suggests using "Anti-N" instead of "Anti N".

R2A6 Anti-N replaces Anti N as follows

"Participants with COVID-19 Anti-N IgG (S/C index) above 1.4 were…"

R2Q7 Comments.

  1. Line 122 suggests using "t-Test" instead of "t Test"

R2A7 t-Test replaces t Test

"…using Independent Samples t-Test…"

Reviewer 3 Report (New Reviewer)

In this paper, the authors have analyzed the presence of a significant levels of antibodies against SARS-CoV-2 in a cohort of 243 cancer patients after 3 or 4 doses of vaccine. They observed that there was an increase in seropositivity of patients after the third and fourth dose. 

- Major comments:

- A better description of the inclusion of patients is mandatory. Initially, I thought it was a longitudinal study, but detection of antibodies was performed only in 142 patients after the second injection. Therefore I assume that these are different patients who were included after a variable number of vaccinations. Therefore, it is essential to explain how these patients were chosen, especially since the number of patients who received a 4th injection is small. The fact that this is not a longitudinal follow-up undoubtedly explains that the antibody titer observed in patients called "Non chemo treated" seems lower (no statistical analysis, unfortunately) after the 3rd injection than after the second.

- the short time between the fourth injection and antibodies detection should be explained. It can artificially increase the percentage of positivity.

- Antibodies detected by the SARS-CoV2-2 IgGII assay does not detect neutralizing antibodies, in contrast to the assay cPass™ SARS-CoV-2 NAb Detection Kit used in ref 71 (Zagouri et al). therefore the sentence which was added page 3, line 102, is irrelevant. I would add that 8 days after my third vaccination, my titer of antibodies, using this assay, was > 40000, but I caught COVID the following weekend (:). In addiction, the results of this study can not be compared with the study of Zagouri et al (discussion p 9, line 344). 

- The classification of patients in 2 groups "Chemotherapy treatment" and "non Chemo therapy treatement"  is not relevant. Administration of medicines constitute a chemotherapy, independantly of  cancer pathologies. It would be better to discriminate "patients with conventional cytotoxic chemotherapy" and patients treated by "targeted therapies".

- The discussion is too long, the comparison with other studies should be more concise.

Minor comment:

page 4, line 152: typo in "receiving". 

Author Response

Please find attached the reply file

Round 2

Reviewer 3 Report (New Reviewer)

Dear authors,

Thank you for your responses to my comments, but I am not convinced by your comment concerning the inclusion of the patients. Rest assured, I know how to read but the phrase you repeat "inclusion criteria..." does not seem sufficient to me. It seems essential to me to clarify that this is not a longitudinal study. I do not agree with this answer "One of the reasons is that some patients did not survive and died between the 3rd and 4th vaccination administration due to their illness." This could have been an evaluable answer if it had been a longitudinal study, but this is not the case. It seems important to me that the fact that this is not a longitudinal study, which is a certain weakness of the study, is clearly indicated and discussed.

Author Response

Ms. Vanessa Hu, [email protected]

Assistant Editor - Diseases                                                                                            September 20, 2023     

 Re: Diseases-2597313  re-submission  

Dear Ms. Hu,

Thank you for considering our manuscript "Response rate to the 3rd and 4th doses of BNT162b2 vaccine in cancer patients administered with active anti-neoplastic treatments" for publication in Diseases.

We thank the reviewer for the thoughtful comments and the helpful constructive feedback, giving us the opportunity to improve and revise the paper.

The reviewer's recommendations have now been incorporated into the manuscript and are documented in the following revision note.

Enclosed please find the response to the referee's comment and the revised manuscript with revisions highlighted in light blue as additions /REVIEW MODE.

We hope that the revised manuscript will fulfill the reviewer comment to satisfaction.

Sincerely,Dr. Abed Agbarya

Review Report Form 3Comments and Suggestions for Authors

Dear authors,

Thank you for your responses to my comments, but I am not convinced by your comment concerning the inclusion of the patients. Rest assured, I know how to read but the phrase you repeat "inclusion criteria..." does not seem sufficient to me. It seems essential to me to clarify that this is not a longitudinal study. I do not agree with this answer "One of the reasons is that some patients did not survive and died between the 3rd and 4th vaccination administration due to their illness." This could have been an evaluable answer if it had been a longitudinal study, but this is not the case. It seems important to me that the fact that this is not a longitudinal study, which is a certain weakness of the study, is clearly indicated and discussed.

 Reply: Thank you for your valuable comment and we understand the importance of the fact that the study is not longitudinal and we did our best to reflect this issue in the manuscript as a limitation to this study.We have added to the discussion section lines 378-388, an indication that the study was not a longitudinal study.

"An important limitation of the current study could be attributed to the fact that it was not a longitudinal study rather a " snapshot" / "status" presentation [71] of the oncology patients' antibodies response to COVID-19 vaccine while receiving active antineoplastic treatment at the time.  

This type of study design is attributed to the fact that in most cases it's difficult to continue a follow-up of cancer patients while on anti-neoplastic treatment for a long period for several reasons such as that some of the patients died, or stopped treatment due to the fact they get cured or sometimes the decision of symptomatic treatment only due to general deterioration and end-stage disease or lost from follow-up. Yet, Figure 1a and Table 2 indicate that the antibody response to the fourth vaccine dose was higher than for the third vaccine dose despite that this is not a longitudinal study."

Of note, during the 2nd vaccine administration, the oncology patients cohort was compared to healthy participants, while during the 3rd and 4th vaccine doses the patients were sorted by their treatment.

In the current study it was difficult to follow up with the vaccinated patients due to the fact that the 2nd, 3rd, and 4th BNT162b2 vaccine doses were administered over a stretch of many months in two separate medical centers, and each individual patient was in charge of scheduling the vaccine appointment at their convenience, moreover some patients drop out. Another reason for a smaller number of participants in the later vaccine doses could be because some patients could have been on remission or "research attrition" and were not consenting to further blood draws. In addition, although the COVID-19 vaccines were offered to oncology patients it does not necessarily mean that all of the patients adhered to recommended guidelines to up take the additional injection.

Although it is possible that some patients included in the study were on active anti neoplastic treatment during more than vaccine administration, it depends on the individual's protocol and therapy regimen.

Reference 71 Longitudinal Study - an overview | ScienceDirect Topics (Accessed 20 September 2023)

We have also make one change in line 75 in the Inclusion criteria update is detailed in lines 73-76:

"Inclusion criteria were ≥18 years old oncology patients, bearing solid tumors who were receiving active antineoplastic treatment when the second, or third or fourth dose of the BNT162b2 vaccine (Pfizer-BioNTech, Kalamazoo, MI, USA) was administered. ".

We hope these changes could answer your concerns.

Best

Abed Agbarya

Round 3

Reviewer 3 Report (New Reviewer)

Thank you for your answers and modifications

This manuscript is a resubmission of an earlier submission. The following is a list of the peer review reports and author responses from that submission.

Round 1

Reviewer 1 Report

This is a report in small number of cancer patients (small in terms of different cancer types) where the authors describe the efficacy of  3rd and 4th vaccine dose in the development of antibodies against SARS-CoV-2.

The main problem of the study is that it is mainly confirmatory. All these results have been previously published several times.

The second major issue is that the authors have tested antibodies against the Wuhan strain of the virus which is no more in the community. Thus, it would be interesting if the authors describe the detection of antibodies against omicron subvariants, like XBB, XBB.1.5, XBB.1.16, BQ.1.1, etc.

Several publications have to be included and discussed in the discussion:

1. Liatsou et al. Cancers (Basel). 2023 Apr 12;15(8):2266.

2. Rosati et al. Cancers (Basel). 2022 Nov 25;14(23):5816.

3. Alexopoulos et al. Eur J Intern Med. 2023 Jan;107:7-16.

4. Zagouri et al. Vaccines (Basel). 2022 Sep 5;10(9):1474.

5. Bergamaschi et al. Front Immunol. 2022 May 25;13:899972.

6. Terpos et al. Cancers (Basel). 2022 Jun 4;14(11):2796.  

Thus I generally believe that the paper does not offer in the relevant literature.

Minor editing in the English is recommended

Reviewer 2 Report

The manuscript by Abed Agbarya et al. focuses on the humoral response of cancer patients treated with the COVID-19 vaccine. The study found that patients who received three or four doses of the BNT162b2 vaccine exhibited higher antibody titers compared to those who received only two doses. This suggests that administering additional doses of the BNT162b2 vaccine to cancer patients undergoing anti-neoplastic treatment could improve their immune response and potentially offer enhanced protection against COVID-19. There are several concerns in this version:

1. It is important to note that the study has limitations, including small cohorts and non-uniform timing of sample collection from the participants. To obtain more robust and generalizable results, the authors should consider expanding the cohorts in future studies.

2. In this version, healthy control should also be enrolled or cited. 

3. The title should be refined because the phrase "Efficacy and duration of response" seems not concrete.

4.  Additionally, there are missing error bars in Figure 1b to ensure accurate data representation.